# Distribution of trespass cannabis cultivation and its risk to sensitive forest predators in California and Southern Oregon

Greta M. Wengert[1]*, J. Mark Higley[2], Mourad W. Gabriel[1,3,4], Heather Rustigian-Romsos[5], Wayne D. Spencer[5], Deana L. Clifford[6], Craig Thompson[7]

1 Integral Ecology Research Center, Blue Lake, California, United States of America, 2 Wildlife Department, Hoopa Tribal Forestry, Hoopa, California, United States of America, 3 United States Forest Service, Law Enforcement and Investigations, Vallejo, California, United States of America, 4 University of California Davis, One Health Institute, Wildlife Health Center, Davis, California, United States of America, 5 Conservation Biology Institute, Corvallis, Oregon, United States of America, 6 Wildlife Investigations Laboratory, California Department of Fish and Wildlife, Rancho Cordova, California, United States of America, 7 United States Forest Service, Missoula, Montana, United States of America

* gwengert@iercecology.org

**Data Availability Statement:** Data on locations of cannabis cultivation sites used to build this model cannot be shared publicly because they are law enforcement sensitive data. Thus, there are strict

## Abstract

Illegal cannabis cultivation on public lands has emerged as a major threat to wildlife in California and southern Oregon due to the rampant use of pesticides, habitat destruction, and water diversions associated with trespass grow sites. The spatial distribution of cultivation sites, and the factors influencing where they are placed, remain largely unknown due to covert siting practices and limited surveillance funding. We obtained cannabis grow-site locality data from law enforcement agencies and used them to model the potential distribution of cultivation sites in forested regions of California and southern Oregon using maximum entropy (MaxEnt) methods. We mapped the likely distribution of trespass cannabis cultivation sites and identified environmental variables influencing where growers establish their plots to better understand the cumulative impacts of trespass cannabis cultivation on wildlife. We overlaid the resulting grow-site risk maps with habitat distribution maps for three forest species of conservation concern: Pacific fisher (*Pekania pennanti*), Humboldt marten (*Martes caurina humboldtensis*), and northern spotted owl (*Strix occidentalis caurina*). Results indicate that cannabis cultivation is fairly predictably distributed on public lands in low to mid-elevation (~800-1600m) forests and on moderate slopes (~30–60%). Somewhat paradoxically, results also suggest that growers either preferred sites inside of recently disturbed vegetation (especially those burned 8–12 years prior to cultivation) or well outside (>500m) of recent disturbance, perhaps indicating avoidance of open edges. We ground-truthed the model by surveying randomly selected stream courses for cultivation site presence in subsets of the modeling region and found previously undiscovered sites mostly within areas with predicted high likelihood of grow-site occurrence. Moderate to high-likelihood areas of trespass cultivation overlapped with 40 to 48% of modeled habitats of the three sensitive species. For the endangered southern Sierra Nevada fisher population, moderate-high likelihood growing areas overlapped with over 37% of modeled fisher denning habitat and with 100% of annual female fisher home ranges (mean overlap = 48.0% ± 27.0

legal restrictions to sharing these data. The data were provided to Integral Ecology Research Center by United States Forest Service Law Enforcement and Investigations with strict restrictions on not sharing those data (data are on loan to IERC but are "owned" by them). To request the data used for this study, researchers who meet the criteria for access to confidential data will need to contact Region 5 US Forest Service Law Enforcement and Investigations, point of contact, Special Agent in Charge, Don Hoang at don.hoang@usda.gov.

**Funding:** This study was funded by United States Fish and Wildlife Service, Yreka Field Office, through a grant administered by National Fish and Wildlife Foundation, grant #0206.15.050267, and funding from California Department of Fish and Wildlife, Endangered Species Act Traditional Section 6 grant #P1482006.The funders had no role in study design, data collection and analysis, decision to publish, or preparation of the manuscript.

**Competing interests:** The authors have declared that no competing interests exist.

SD; n = 134) in two intensively studied populations on the Sierra National Forest. Locating and reclaiming contaminated cannabis grow sites by removing all environmental contaminants should be a high priority for resource managers.

## Introduction

Evidence of wildlife poisonings, habitat destruction, and pesticide contamination associated with trespass cannabis cultivation on public lands has quickly accumulated in forested regions of California and southern Oregon [1–4] since discovery in 2009 that anticoagulant rodenticides used at grow sites were contributing to fisher (*Pekania pennanti*) mortalities in California [5]. However, the true extent of the problem is poorly understood, because only a fraction of trespass cultivation sites is thought to be discovered by law enforcement. Further, only a small portion of the sites found by law enforcement is remediated, and those that are un-remediated and those as yet undiscovered continue to pose environmental threats on our public lands.

Two species listed under the federal Endangered Species Act that appear to be at risk from the effects of this activity are the endangered fisher [5–7] and the threatened northern spotted owl (*Strix occidentalis caurina*, [2, 8]), for which direct mortality as well as sublethal effects from pesticides used at these sites are well documented. Both species occur throughout overlapping as well as separate forested regions of southwestern Oregon, northwestern California, and the southern Sierra Nevada [9, 10] and are strongly associated with mid-elevation coniferous forests where many trespass grow sites have been found. Consequently, risks from anticoagulant rodenticides found at trespass cannabis sites was a factor in listing fishers in the Sierra Nevada range in California as endangered [11]. In addition, coastal Oregon and California populations of the Pacific marten (*Martes caurina)*, just recently listed as federally threatened, inhabits forests in coastal California and Oregon [12, 13] where grow sites are also common. All three sensitive forest species consume prey at risk of contamination with anticoagulant rodenticides and other toxic pesticides within cultivation sites [9, 10, 14].

Eradication and reclamation (elimination of trash, infrastructure, and chemicals), followed by ecological restoration, are necessary to mitigate any threats to the environment, wildlife and other resources from trespass cannabis cultivation sites. However, financial and logistical challenges limit the ability of law enforcement to locate trespass grow sites using aerial reconnaissance or other methods, and discovery of all sites is infeasible. In addition, fully evaluating and addressing the cumulative impacts from cannabis cultivation throughout the region requires an understanding of how cannabis cultivation sites are distributed across the landscape. A systematic approach for mapping the likely distribution of trespass grow sites is therefore needed to identify where the impacts could be most severe and to facilitate more efficient and targeted grow-site discovery, reclamation, and remediation.

We know of two previous efforts to model cannabis cultivation in California. In Humboldt County, California, Butsic et al. [15] modeled the distribution of private land cultivation sites and found that they are most likely to exist near other cultivation sites but were not predicted by other environmental variables the study examined. That study was limited to cultivation by landowners on their own properties, which differs drastically from trespass cultivation on public land, where a primary goal of the cultivators is evading detection by law enforcement long enough to harvest the cannabis and ensure a healthy profit. Koch et al. [16] modeled socio-economic and environmental drivers of cannabis cultivation on national forests across California, Oregon, and Washington, using predictors with fine to very coarse resolutions (from 10m to

an entire state). That study considered a small number of environmental (terrain, water, and climate) variables and did not include measures of vegetation cover or other ecological factors likely also important to grow-site selection. More accurate maps at relatively fine resolution (e.g., sub-watershed scale) based on a more complete set of predictive environmental variables would further our understanding of the potential impacts of trespass grow sites on wildlife and would help law enforcement and resource managers locate and remediate sites on public lands.

A statistical modeling framework typically applied to modeling species and habitat distributions, such as ecological niche-factor modeling, could be an effective approach to analyzing and mapping grow-site distribution such that the risks to these three species can be extrapolated on a landscape scale. Rather than identifying environmental factors affecting species distributions, we applied the approach to predict where cannabis growers select sites for cultivation. Specifically, we used maximum entropy (MaxEnt) modeling [17] to predict the likely distribution of cannabis grow sites in forested regions of California and southern Oregon to better (1) understand the spatial extent of associated environmental impacts, (2) understand the potential distribution of population-level effects of associated environmental damage to three species of high conservation concern, and (3) provide law enforcement and resource managers with information to help prioritize and make more efficient their efforts to discover, eradicate, and remediate such sites.

## Methods

### Cultivation site distribution modeling

We received data from local, state, and federal law enforcement agencies on the locations of 1469 trespass cannabis cultivation sites on public and nearby private and tribal lands in forested regions of California and southwestern Oregon from 2007 through 2014 (Fig 1). We only received data on cultivation site presence, and had no information on where law enforcement agents searched for, but did not find, cultivation sites (absence). We therefore used the presence-only MaxEnt modeling system [17–19]; MaxEnt model version 3.3.3k: https://biodiversityinformatics.amnh.org/open_source/maxent/) to model grow-site distribution. MaxEnt is often used to model species distributions (sometimes interpreted as habitat quality) using species presence data and the values of environmental variables at these locations relative to random background points. Our study differed from typical distribution modeling in that our goal was to model human choice, which we assumed was dictated by a combination of environmental, climatic, and anthropogenic variables.

We filtered the cultivation site data using a 2000m minimum nearest-neighbor distance to avoid inflating the representation of sites that may have been part of the same cultivation complex and to reduce model overfitting due to biased sampling by law enforcement during reconnaissance efforts [20]. Filtering reduced the 1469 sites to 945 presumably independent locations. For background (availability) points, we used a random selection of 10,000 points across the modeling extent, because we did not have information on the areas searched by law enforcement and thus we were unable to limit background data selection to surveyed areas only [21].

Initially, we divided the modeling extent into five sub-regions using ecoregional boundaries [22] on the assumption that growers might use different selection criteria in regions differing significantly in climate, vegetation, or other factors (Fig 1). We compared the single, seamless model obtained by modeling all five regions together (i.e., our entire modeling extent) to outputs obtained using separate sub-regional models, and determined that modeling the entire extent produced a better-fit model and a more parsimonious map of risk than the region by

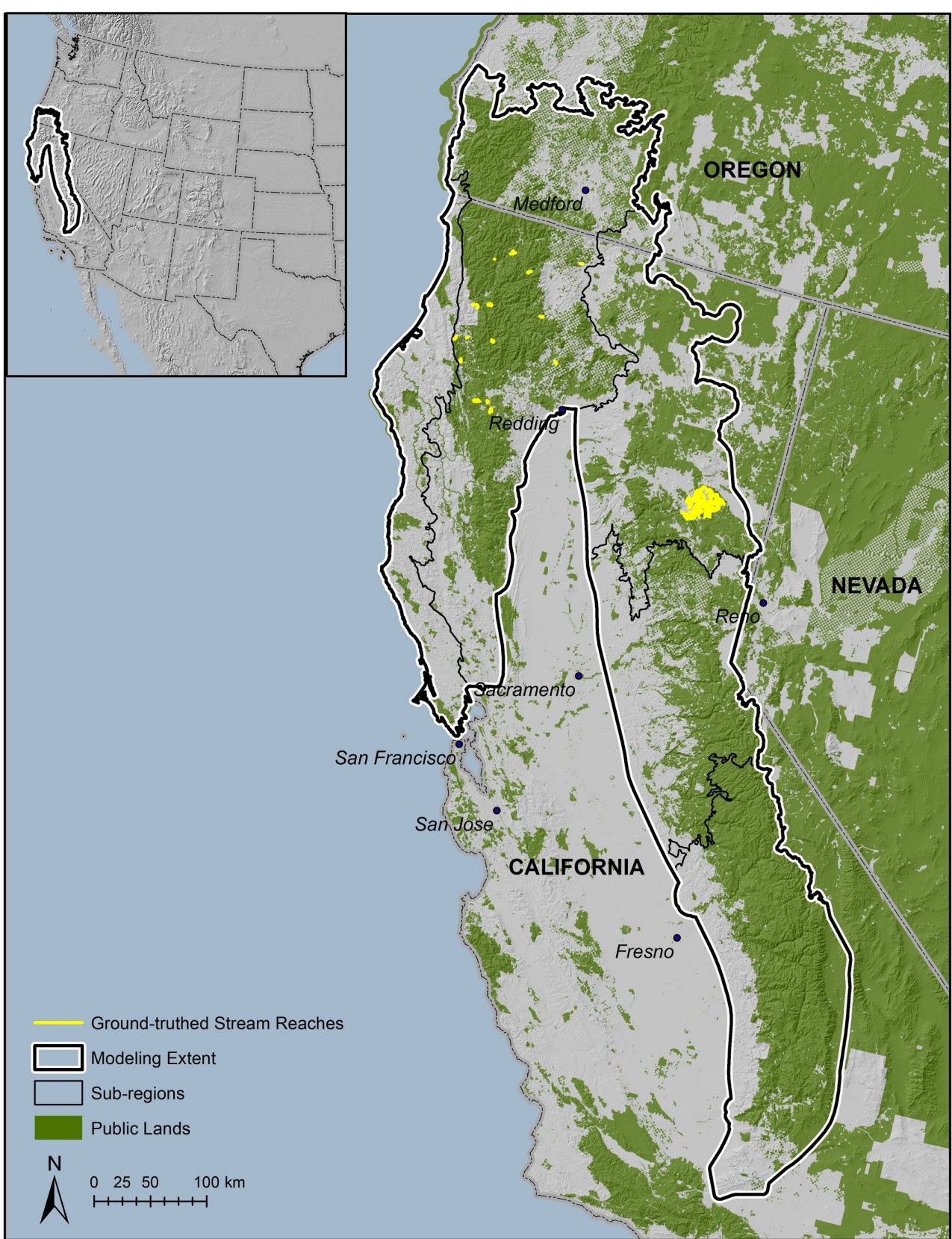

**Fig 1. Study area showing modeling extent and five ecoregional subdivisions used for MaxEnt modeling of trespass cannabis grow sites, and subareas subject to ground-truthing.**

region approach. We therefore used the single, whole-extent model for all subsequent analyses.

We selected candidate environmental variables for modeling at 90m resolution based on personal observations and discussions with law enforcement (Table 1). In addition to variables derived from climate, terrain, and land cover data, we used an array of variables related to human influences, such as distance from roads, populated areas, and ownership boundaries. Because our observations suggested that growers often select forest stands that had been logged or burned in recent years (typically within the prior 2 decades), we also tested several variables derived from recent disturbance data layers (4–12 or 8–12 years prior to cultivation year). We averaged variables over a 450m radius moving window to approximate the resolution of cultivation site locality data and average footprint (based on discussions with law enforcement and collected data). Ninety percent of the filtered cultivation site locations were used for model training, and 10% as test data. We implemented 10-fold cross validated replication with 500

**Table 1. Candidate environmental variables evaluated for use in the MaxEnt cannabis grow-site distribution model at 90m resolution.**

| Variable | Source |
|---|---|
| Mean basal area weighted stand age | GNN Forest Structure 2012, http://lemma.forestry.oregonstate.edu/data/species-maps |
| Mean tree canopy cover (%) | USGS National Land Cover Database (NLCD) Tree Canopy Cover 2011 |
| Mean tassel-capped greenness | CBI/LANDSAT TM circa 2000 |
| Mean tassel-capped wetness | CBI/LANDSAT TM circa 2000 |
| Mean distance to nearest disturbance (4–12 years previous to 2014) | LandTrendr (Landsat-based Detection of Trends in Disturbance and Recovery), http://geotrendr.ceoas.oregonstate.edu/landtrendr/ |
| Proportion disturbed (4–12 years previous to 2014) | LandTrendr (Landsat-based Detection of Trends in Disturbance and Recovery), http://geotrendr.ceoas.oregonstate.edu/landtrendr/ |
| Mean distance to nearest disturbance (8–12 years previous to 2014) | LandTrendr (Landsat-based Detection of Trends in Disturbance and Recovery), http://geotrendr.ceoas.oregonstate.edu/landtrendr/ |
| Proportion disturbed (8–12 years previous to 2014) | LandTrendr (Landsat-based Detection of Trends in Disturbance and Recovery), http://geotrendr.ceoas.oregonstate.edu/landtrendr/ |
| Mean latitude adjusted elevation | USGS National Elevation Dataset (NED; 1 arc-second) |
| Mean solar insolation index | Derived from USGS National Elevation Dataset (NED; 1 arc-second) |
| Mean percent slope | Derived from USGS National Elevation Dataset (NED; 1 arc-second) |
| Standard deviation percent slope | Derived from USGS National Elevation Dataset (NED; 1 arc-second) |
| Mean value of transformed slope aspect | Derived from USGS National Elevation Dataset (NED; 1 arc-second) |
| Average daily maximum temperature July 1981–2010 | PRISM Climate Group, Oregon State University, http://prism.oregonstate.edu |
| Mean annual precipitation | PRISM Climate Group, Oregon State University, http://prism.oregonstate.edu |
| Proportion in public conserved land | Protected Areas Database of the United States (PAD-US, CBI Edition) 2012 |
| Mean distance to nearest private lands | Protected Areas Database of the United States (PAD-US, CBI Edition) 2012 |
| Mean distance to nearest fresh water | USGS National Hydrography Dataset (NHD), medium resolution |
| Mean distance to nearest perennial fresh water | USGS National Hydrography Dataset (NHD), medium resolution |
| Mean distance to nearest road | ESRI, NPS, USFS |
| Mean distance to major road | ESRI |
| Mean distance to nearest populated place | US Geological Survey, Geonames, https://www.usgs.gov/core-science-systems/national-geospatial-program/geographic-names |

iterations. Area under the receiver operating curve (AUC) was used for model evaluation, ranging from 0.5 (random) to 1.0 (perfect discrimination). For apparently related groups of variables, we ran default univariate models using 10-fold cross-validation. For each group of correlated variables ($|r| > 0.7$), we selected the variable with the highest mean cross-validation test AUC for inclusion in the full multivariate model and excluded all others in the group.

Starting with the full model, we used step-wise variable reduction (jackknife approach, eliminating the variable causing the smallest decrease or largest increase in mean test AUC when removed) to assess the importance of variables in the final model. We retained the model with the highest mean test AUC. We tested feature types (auto-features, the default; linear/quad/product; hinge only) and regularization parameters (0.5, 1 (the default), 1.5, 2, 3, 5) to avoid overfitting using the selected model (18 combinations tested) selecting the feature type and regularization parameter combination yielding the minimum AICc (Akaike information criterion corrected for small sample sizes) with ENMTools [23]. Once we determined the most appropriate feature types and regularization parameters, we reran the selected model with those settings. Final logistic output of the model predictions was classified into low, moderate, and high likelihoods using Boyce Index Analysis [24] and exported to ArcGIS 9.1 for further analysis and mapping. In addition to mapping the classified results across the modeling extent, we also summarized them by HUC12 watershed units as the proportion of each unit in moderate to high grow-site likelihood. Mapping the results this way illustrates how the model outputs could be used to help prioritize surveillance and remediation efforts at a useful landscape scale.

## Model validation

Because of the spatial bias inherent to the grow-site locality data due to non-random sampling by law enforcement (they typically search at and near previously found cultivation sites or where tips or other observations lead them) we wanted to ensure our model accounted for characteristics of cultivation sites not previously found by law enforcement as well as those that were. We therefore used an unbiased method of finding cultivation sites previously undetected by law enforcement by ground-truthing randomly selected drainages within two subsets of our overall modeling extent during 2015–2017. The subsets were chosen based on logistical feasibility and available funding, thus limiting sampling to the Klamath Basin in northwestern California and portions of Plumas National Forest in the northern Sierra Nevada (Fig 1).

To select streams for ground-truthing, we overlaid our classified final model with the CalWater coverage (https://gispublic.waterboards.ca.gov/arcgis/rest/services/webmap/CalWater_HydrologicAreas/MapServer/0) for the Klamath Basin and selected portion of Plumas National Forest. Our objective was to perform ground-truth surveys on 1% of total stream distance within each ground-truthing area. In the Klamath Basin, we did this by randomly selecting a single reach, then selecting all reaches within that HUC12 subwatershed with the same name, and adding the total stream mileage until we achieved 1% of the entire CalWater stream distance within each ground-truthing area. On the Plumas National Forest, we used survey data from a separate, unrelated project which included every stream reach within that project area. This resulted in the selection of 16 streams (398 stream km) in the Klamath Basin, of which we were able to survey 178 km, and 73 streams (709 km) on the Plumas National Forest, all of which were surveyed.

Because all cultivation sites require irrigation, irrigation pipe always originates from some natural or modified water source, usually within a few hundred meters upslope of the cultivation site, but up to 3 km away according to our observations. We therefore performed ground-truthing by walking stream courses searching for irrigation pipe, water cisterns, refuse typically

associated with trespass cultivation (e.g., fertilizer or pesticide containers, propane tanks, sleeping bags, tents, and other camping materials), or cannabis planting holes themselves. We generally only performed surveys during the non-growing season (November–early April) for safety reasons unless surveyors were accompanied by law enforcement personnel. We described any evidence of cultivation we found and recorded a location with GPS. When possible, we identified the precise location of the cultivation plots by following the irrigation pipe to its final destination. When we found other types of evidence but not irrigation pipe, we searched the immediate area for the actual location of the cultivation plot or other site features.

We also tested the model with an independent data set obtained from law enforcement agencies for a large subset of cultivation site locations (n = 92) from 2015 and 2016 to determine whether the model built on location data 2007–2014 accurately predicted spatial distribution of cultivation site likelihood in later years. This would test for potentially changing trends in site selection by growers.

We calculated the percentage of cultivation sites found both via ground-truthing and the 2015–16 dataset within each likelihood category in the model to assess how well the model fit to independent cultivation site location data. For the ground-truthing results, we used Chi Square in an observed vs. expected framework using the likelihood categories (high, moderate, low), where expected numbers of grows in each likelihood category equaled the proportion of stream kilometers surveyed in each category multiplied by the total number of grows discovered.

## Overlap of grow-site likelihood with species of concern

We acquired habitat models for three species of concern in the analysis area: northern spotted owl, Humboldt marten, and Pacific fisher. A model of "high quality habitat" for northern spotted owl was obtained from Davis et al. [10]. For Humboldt marten, a habitat model was not available at the time of our analysis, so we used a recently published model of marten habitat "core areas," which were defined as "likely to contain sufficient high quality habitat to support long-term occupancy by coastal martens. . ." [25]. For fisher, we used a thresholded MaxEnt model of "selected-for" (hereafter, suitable) and "selected-against" (unsuitable) habitat throughout its range in California and southern Oregon [26]. For the recently listed endangered Distinct Population of fisher in the southern Sierra Nevada, we also overlaid a finer-resolution fisher denning habitat model [9] and female fisher annual home range polygons to assess potential population-level effects of grow-site likelihood on this isolated population. The denning and home range data were made possible by two intensive, multi-year telemetry studies of fishers on Sierra National Forest [27].

The denning model was created using MaxEnt, 350 natal and maternal dens identified from 2008–2013, and environmental variables averaged at 2km resolution [9]. Annual home ranges (n = 134) were calculated for adult female fishers monitored on the Sierra National Forest between 2007 and 2018. Because monitoring techniques varied across research projects and years, we calculated home ranges using the nonparametric local convex hull approach (LoCoH: [28]. We used the adaptive LoCoH function in the R package T-LoCoH [29], and calculated 95% isopleths for animals with >25 locations. This method has proven to be superior to more traditional home range estimation techniques, such as minimum convex polygon or fixed kernel analysis, when dealing with animals that have distinct territorial boundaries, are sensitive to edge habitat, or when the data includes a variety of monitoring methods [30, 31]. Location data used included aerial, ground, and GPS telemetry, as well as rest and den sites. Additional information on capture techniques and monitoring methods can be found in Sweitzer et al. [27] and Green et al. [32]. We overlaid moderate-high cultivation site likelihood

areas generated by our model with the various habitat layers and the fisher home ranges and calculated percent of predicted suitable habitat and fisher home ranges that were intersected by high-moderate cultivation site likelihood.

## Results

### Modeling results

The final pruned, tuned, and thresholded model (Fig 2) predicts that there are 21,607 km$^2$ of high likelihood grow-site potential in the study area (13% of the model extent), 24,300 km$^2$ of moderate likelihood (14%), and 125,421km$^2$ of low likelihood (73%). Fig 3 maps the proportion of HUC6 watershed units in the moderate to high likelihood categories to illustrate how the model results can be presented for assessing resource impacts or helping prioritize surveillance and remediation efforts at that scale.

After running the univariate analyses on all groups of related variables, 11 variables advanced to the model-pruning stage: mean basal-area weighted stand age, mean tree canopy cover, mean distance to nearest disturbance (8–12 years prior to 2014), mean latitude-adjusted elevation, mean percent slope, mean distance to nearest private lands, mean distance to nearest fresh water, mean distance to nearest road, mean distance to nearest populated place, mean value of transformed slope aspect, and average annual precipitation. Through stepwise variable reduction, mean distance to nearest populated place was removed while all remaining predictors retained a percent contribution level of at least 1%. Mean percent slope and mean latitude-adjusted elevation had the greatest importance, but measures of vegetation structure and disturbance, climate, water, and human land uses also contributed (Table 2). The best combination of feature type and regularization multiplier was the default parameters (auto-features and 1.0). The model performed satisfactorily with an overall mean test AUC of 0.8, and did not suffer from overfitting, with a mean 10% omission rate of 0.125.

Fig 4 (univariate response curves) illustrates how each variable alone relates to predicted grow-site likelihood. Interpreting these (in concert with marginal response curves and variable maps, not shown) reveals that trespass grow sites are generally associated with moderate to steep slopes (~30–60%), at ~800-1600m elevation, close to private land (<2.5km), with >50% tree canopy cover, ~1–2 km from a road, and within ~200m of water, over a broad range of annual precipitation (~70-300cm/year), in relatively young forest stands (~50–120 years), with moderate basal area of live trees (~20-80m$^2$/ha). The results also suggest that growers preferentially site grow plots either inside areas recently (8–12 years prior) disturbed by fire or logging, or more than 500m outside of recent disturbance.

### Model validation

Our ground-truthing surveys during 2015–2017 located current or historical evidence of cultivation infrastructure at 16 sites previously undetected by law enforcement along 11 independent streams (out of 89 total streams surveyed; 12%). We were able to identify the actual cultivation plots at 14 of these 16 sites. Cultivation plot location was significantly associated with likelihood category (p < 0.0001), with 12 of 14 plots (86%) falling into either the high (9) or moderate (3) likelihood category, suggesting our model was quite successful at identifying areas with a high likelihood of cultivation that had escaped detection by law enforcement. The two plots that fell into the low likelihood category were within 500 m of predicted high or moderate likelihood. Of 92 cultivation sites found by law enforcement in 2015 and 2016, 34 (37%) fell into the high likelihood category, 26 (28%) in moderate, and 32 (35%) in low. For those that did not fall in the high likelihood category, the mean distance from the site to high likelihood grow habitat was 740 m, with a range of 6–4705 m.

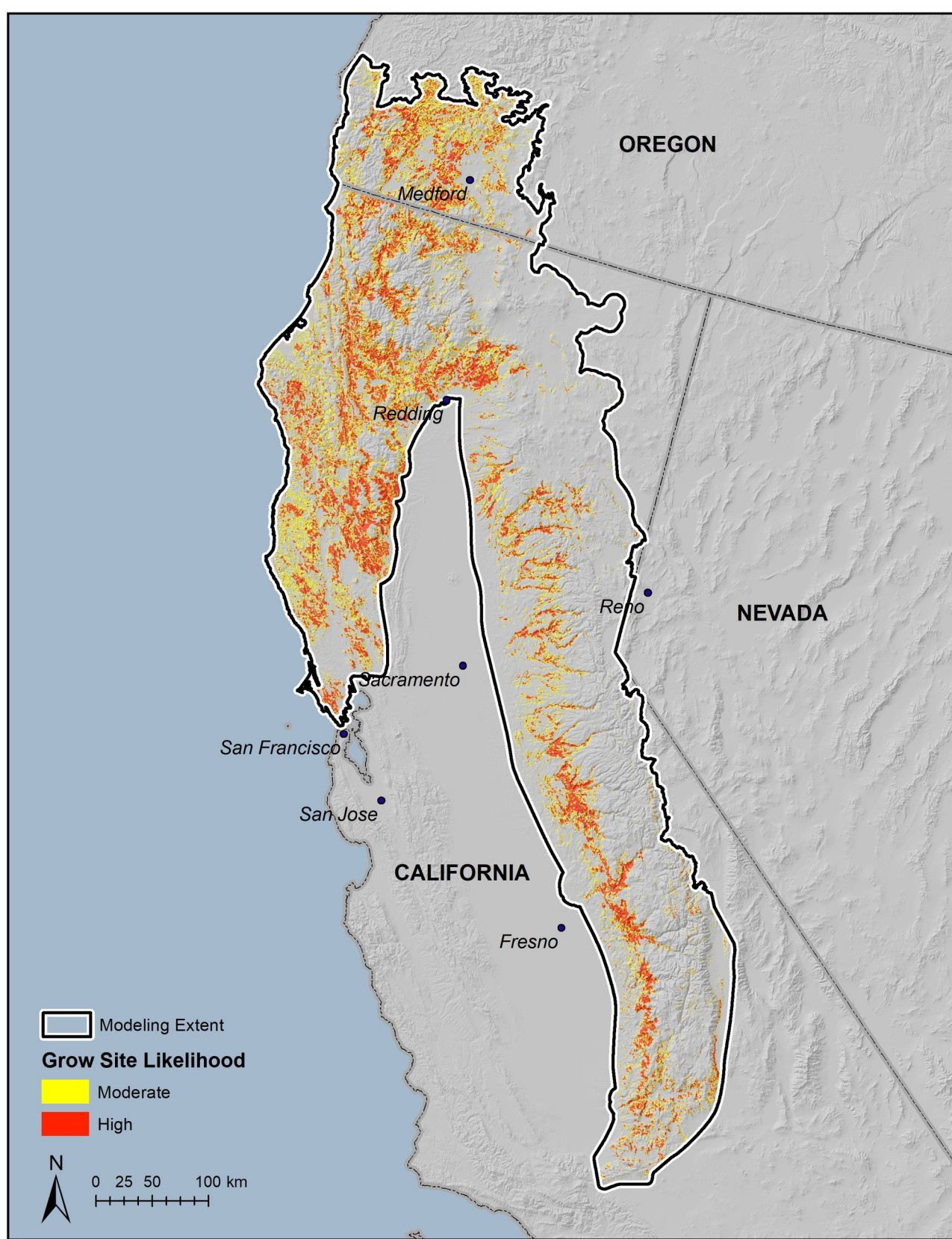

**Fig 2. Modeled cannabis cultivation likelihood using MaxEnt and locality data from 2008–2014.**

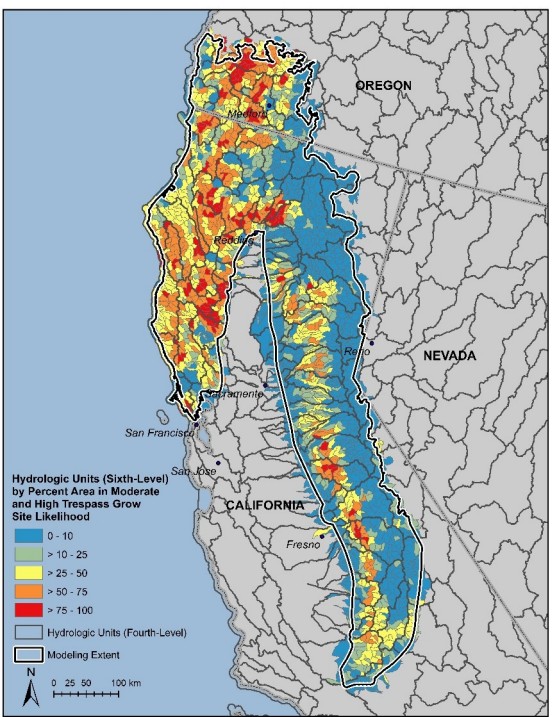

**Fig 3. Proportion of modeled moderate-high cannabis cultivation likelihood summarized within HUC12 subwatersheds in the modeling extent.**

## Overlap of cultivation site likelihood with species of concern

High and moderate cultivation site likelihood overlapped with 44.4% of modeled fisher habitat; 47.9% of highly suitable spotted owl habitat; and 39.9% of Humboldt marten habitat cores (Fig 5). For the two distinct portions of the occupied fisher range, in northwestern California and Oregon and in the southern Sierra Nevada, fisher habitat overlapped with moderate to high cultivation likelihood by 53.6% in the north but only 22.0% in the south.

Because the highest quality fisher denning habitat also tends to be concentrated in lower elevation portions of the fisher habitat band, where oaks intermix with conifers [9], overlap of denning habitat with grow-site potential rises to 37.7% within the southern Sierra Nevada.

**Table 2. Contribution and importance of each predictor variable for the final model estimating relative likelihood of trespass cannabis cultivation in forested areas of California and southern Oregon 2008–2014.**

| Variable | Percent contribution | Permutation importance |
|---|---|---|
| Mean percent slope | 31.0 | 20.6 |
| Mean latitude adjusted elevation | 24.9 | 31.3 |
| Mean distance to nearest disturbance (8–12 years previous) | 11.5 | 6.2 |
| Mean distance to nearest private lands | 9.2 | 10.2 |
| Mean tree canopy cover (%) | 7.6 | 11.5 |
| Mean distance to nearest road | 6.0 | 7.3 |
| Mean distance to nearest fresh water | 3.5 | 3.9 |
| Mean annual precipitation | 3.4 | 5.3 |
| Mean basal area weighted stand age | 1.9 | 2.3 |
| Mean value of transformed slope aspect | 1.0 | 1.3 |

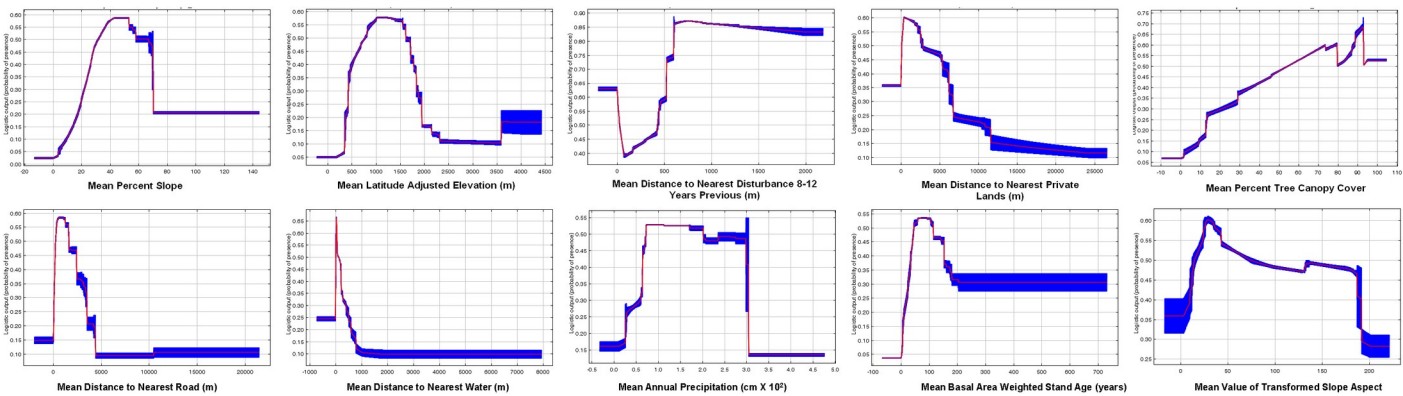

**Fig 4. Univariate response curves for all predictor variables selected for the final model of trespass cannabis cultivation site likelihood in the modeling extent.**

**Fig 5. Overlap of moderate to high trespass cannabis cultivation site likelihood (red) with modeled habitat for Pacific fisher (left), northern spotted owl (center) and Humboldt marten (right).**

Finally, of 134 annual female home ranges in the Sierra Nevada, all (100%) had at least some overlap with moderate-high grow-site potential. The areal extent of overlap varied from ~1% to 100% of each home range, with a mean overlap of 48.0% $\pm$ 27.0 SD.

## Discussion

This unique collaboration between law enforcement and wildlife biologists, which applied a species distribution modeling technique to human choice, provides valuable insights for both law enforcement and resource managers. Our cannabis cultivation distribution model, built using MaxEnt methods and data provided by law enforcement, predicts where illegal trespass cannabis cultivation is likely having the greatest impacts on wildlife and other resources on public lands in California and southern Oregon. Intersecting the model results with habitat distribution models for three forest species of conservation concern illustrates how pervasive the negative impacts of trespass cultivation may be, including the potential effects of associated pesticide poisoning on populations of endangered species. The modeling approach could be extended with additional data on grow-site abundance and associated pesticide loads to better quantify environmental impacts. Perhaps most important, maps of likely grow site distribution can make efforts to find and remediate trespass cultivation sites more efficient.

### Modeling results

Our model performed better when we modeled the five ecoregions together, rather than when we modeled each region separately. Though unexpected, this result suggests that growers are choosing sites based on similar landscape features throughout the study region, despite significant differences in climate and land covers across the large, ecologically diverse modeling extent. Growers appear to be selecting cultivation sites based on a combination of abiotic, anthropogenic, and vegetation structure factors, and less on climatic conditions or land cover types (e.g., forest composition).

The model results demonstrate that trespass cannabis cultivation is widely and predictably distributed in forested regions of California and southern Oregon in low to mid-elevation (~800-1600m) forests, on moderate slopes (~30–60%), and close to perennial water (generally <200m). Although sites are mostly on public land, they tend to be close to private land (<2.5km from boundaries), which may also reflect a tradeoff between access and detection risk, or indicate a nexus with private land cultivation making use of public land resources.

Forests at grow sites tend to have moderate to dense tree canopy cover (>50%) and moderate basal area of live trees (~50-120m$^2$/ha) but with no evidence that tree species matter (sites are found in a wide array of forest types, from oak woodland to pine to diverse mixed conifer types). Somewhat paradoxically, results also suggest that growers either prefer sites inside of recently disturbed vegetation (especially those burned 8–12 years prior to cultivation) or well outside of (>500m from) recent disturbance. Siting within recently burned areas likely reflects a preference for post-fire vegetation conditions (e.g., sufficient sunlight for cannabis growth but with some brushy understory to obscure plots). This association with recent disturbance was also evident during field surveys. For example, on the Plumas National Forest in 2015 and 2016, 16 cultivation complexes were discovered within the Moonlight Fire footprint, a 263-km$^2$ fire occurring in 2007. In the Trinity Alps Wilderness on the Shasta-Trinity National Forest, 13 cultivation complexes found in 2014–2015 were within fire scars. On the other hand, siting far outside of recently disturbed areas may reflect avoidance of open edges where plots could be more easily discovered; however, this hypothesis deserves further study.

The strong association with mid-elevations probably reflects a trade-off between longer growing season and better climate conditions at lower elevations, and lower human presence

and more public land at higher elevations. Grow sites also tend to be concentrated 1–2 km from roads, which probably facilitates grower access while avoiding places too close to roads (<1km) to minimize risks of detection. The preference for moderately steep slopes near water sources is because all grows use gravity-fed irrigation, which probably helps explain the very broad range of precipitation (~70–300 cm/year) at grow sites. Even steeper slopes (>60%) would make access and cultivation difficult. Although growers require a season-long water source, and most cultivation plots are very close to (<200m from) the source, we have observed irrigation lines running up to several kilometers to grow plots. The statistical association of sites with mapped water sources was actually less strong than we expected, in part because of these sometimes long irrigation lines, and in part because many perennial springs used by growers are missing from available geospatial hydrology data sets, including fairly thorough in-house data sets in U.S. Forest Service district offices. It is unknown how growers find these undocumented springs, although law enforcement officers suspect that early season reconnaissance by growers includes a search for water sources.

## Model validation

The model performed well at predicting where previously undetected sites are likely to be found, as evidenced by our ground-truthing surveys which detected many sites dated from approximately 2007 through 2014. Our model did not perform as well when tested with more recent (2015–16) cultivation site location data. This could indicate model inaccuracies, changes in grower selection criteria over time, changes in law enforcement surveillance methods or success over time, changes in environmental variables used to build the model, or a combination of these influences. Indeed, law enforcement explained that both they and growers have altered their geographic patterns over the years (Commander Jack Nelsen, California Department of Justice, Campaign Against Marijuana Planting Program Director, personal communication): During 2016–17 law enforcement focused their discovery efforts on public, especially federal, lands, which may have caused growers to switch to using private property more frequently. This pattern shifted again in 2018–19, when state agencies increased their enforcement on private lands by newly developed enforcement teams in Northern California, and growers shifted back to using more public lands again. This shift also coincided with an apparent decline in trespass grow sites in northwestern California during 2018–19, and an increase in the southern Sierra Nevada. At the same time, anecdotal field observations suggested that growers shifted from planting large cultivation plots with many plants each in earlier years, to more and smaller clusters of plots with fewer plants each in recent years. This shift to smaller, more dispersed cultivation plots may also alter growers' site selection criteria. For example, smaller plots may need less canopy cover for concealment from aerial detection, or growers may use small breaks in otherwise dense forest canopy, thus weakening the statistical relationships between plot localities and vegetation variables like canopy cover and tree basal area, which are measured at broader scales.

In a previous modeling effort, Koch et al. [16] concluded that while certain environmental variables, such as slope and aspect, influenced where growers located their cultivation sites, the current price of cannabis also influenced their choices. Perhaps cannabis price fluctuations and the recent legalization of recreational cannabis in California and Oregon also contributes to declining accuracy of our model in later years.

In addition to changes in site selection patterns by growers, vegetation structure also shifted dramatically in large portions of the study area over the course of the study. Beginning in 2014, the central and southern portions of our analysis area were hit with a significant wave of drought and bark beetle infestation, which by 2016 had killed approximately 129 million trees,

with mortality rates reaching 90% in some areas [33]. Given the associated changes in tree canopy and light penetration, it is likely that site-specific cannabis growing conditions changed accordingly. It is therefore unsurprising that our model, developed using both plot location and vegetation data collected prior to this event, performed less well when assessing post-2014 locations. It is also likely that the more dispersed approach to establishing grow plots in recent years also dispersed the environmental threats, such as pesticide exposure, more broadly. It would be beneficial to update our model with post-drought vegetation data to ascertain how grow-site distribution may have changed with these vegetation changes. And given continuing changes in conditions and grower siting practices, the model could be made dynamic, with regular (e.g., annual) updating as practices and conditions change and new data become available.

Unfortunately, our ground truthing results, which discovered 16 previously unknown cultivation sites along 887 km of surveyed streams, substantiate the concern that law enforcement detects only a small fraction of the trespass cultivation sites in California and Oregon. In the Klamath Basin alone, we found 11 previously undetected sites by surveying only 0.5% of the available stream length, suggesting that the number of sites remaining to be discovered in this region alone is extraordinarily high.

In some cases, cultivation plots or complexes of associated plots spanned two or even all three predicted likelihood classes. Although the model generally performed well at the chosen 90m resolution, a coarser-resolution model might perform better and be more useful to law enforcement and land managers for prioritizing aerial reconnaissance or remediation efforts. We therefore summarized the likelihood classes by HUC12 subwatersheds (average size ~104 km$^2$; Fig 3) to illustrate an approach that could be replicated at any suitable resolution, such as over entire watersheds, reserve areas, ranger districts, townships, or sections.

## Overlap of cultivation site likelihood with species of concern

The significant degree of overlap of predicted grow-site likelihood with modeled habitat of Pacific fisher, northern spotted owl, and Humboldt marten demonstrates there may be significant risks to these species from illegal cannabis cultivation, especially stemming from toxicant exposure. Over 53% of modeled fisher habitat within its occupied range in the north is predicted to be at moderate to high likelihood of cultivation site occurrence. In the more isolated and endangered distinct population segment of fishers in the southern Sierra Nevada, there was only 22% overlap of suitable habitat and moderate-high cultivation risk, apparently due to the fairly distinctive and overlapping elevation bands of high-risk cultivation landscape (from ~800 to 1600 m elevation) and suitable fisher habitat (~1,220–2740 m) in the southern Sierra Nevada. However, over 37% of denning habitat was at risk, apparently because denning habitat tends to be concentrated in the lower elevations of suitable fisher habitat. Moreover, all reproductive female home ranges (n = 134) overlapped areas of moderate-high grow-site potential by an average of nearly 50% per home range. Thus, all denning female fishers have at least some grow-site potential in their home ranges and likely experience exposure to associated rodenticides, as evidenced by the finding that 82% of fishers in the population have evidence of pesticides in their tissues between 2007 to 2019 [34]. The population-level risk this poses to fishers is probably magnified by timing, because the denning period when kits most depend on the mother (late March-May) coincides with the initiation of cultivation and heavy use of rodenticides [35], which probably increases the risk to fisher reproduction and recruitment. The fact that grow site potential overlapped even more with suitable habitat for the unlisted, northern population of fishers than with the listed southern population suggests that fishers

are being adversely affected by poisoning at grow sites throughout California and southern Oregon. In northern California, 75% of 48 tested fishers had pesticides in their tissues [5].

Fishers are commonly recorded by camera traps at trespass cultivation sites throughout northern California, often interacting with refuse and pesticides there [35]. During a separate study of prey population dynamics at grow sites, 10% of captured rodents within grow sites tested positive for anticoagulant rodenticides, while none in nearby controls (non-grow sites) tested positive [34]. These observations reinforce the evidence that the source for most if not all exposure to and death from anticoagulant rodenticides in fishers, which affects at least 85% of the population, is trespass cultivation sites [5].

While the death of some fishers has been directly attributed to anticoagulant poisoning [5, 7], sublethal anticoagulant exposure may contribute to other proximate causes of mortality [7]. Many species have exhibited lethargy and weakness with sublethal exposure levels [36], potentially making fishers more prone to predation, the leading cause of fisher mortality [37]. Moreover, evidence from camera monitoring at grow sites suggest that both fishers and their primary predators—mountain lion (*Puma concolor*) and bobcat (*Lynx rufus*)—are attracted to these sites, increasing potential for predatory encounters [35, 37]. Thompson et al. [6] linked decreases in female fisher survivorship rates in the Sierra Nevada to the number of known cultivation sites in their home ranges, and we found that moderate-high cultivation site likelihood overlaps with all female fisher home ranges by an average of nearly 50%. These findings reinforce the hypothesis that trespass grow sites are an additive factor directly affecting fisher populations and may also prevent their expansion into apparently suitable but unoccupied habitat [38].

Northern spotted owls are similarly at risk from contamination at grow sites, with 48% of their habitat overlapped by moderate-high likelihood of cultivation. The measured exposure rate of spotted owls to anticoagulant rodenticides is 70%, approaching the high levels measured in fishers [8]. Northern spotted owls continue to face threats of habitat modification and fragmentation from logging, competition from barred owls (*Strix varia*), and now habitat fragmentation [39] and pesticide exposure [8] from cannabis cultivation. Recent evidence suggests that barred owls suffer less exposure to rodenticides than spotted owls [8], further increasing their competitive advantage. Together, these factors threaten the continued viability of northern spotted owl populations in the region [40].

Although we were unable to obtain a habitat suitability map for the Humboldt marten, we did perform a coarse assessment using modeled habitat cores. It is not surprising that the overlap of moderate-high grow-site potential with marten habitat cores was lower than with fisher and northern spotted owl habitat, because martens are strongly associated with old-growth, dense-canopy forest patches or forests on serpentine soils [41], neither of which appear to be selected by growers. However, with over 37% overlap of key habitat core areas with moderate-high grow-site likelihood, and knowledge that martens are also at risk of rodenticides used at cultivation sites to control rodents [14], risks to this threatened species are also significant.

## Conclusions and recommendations

Trespass cannabis cultivation seems to have increased rapidly in the western United States in the past decade, although this may be due to increased awareness of the issue rather than actual increase in trespass cultivation. The sheer scope of the problem and the large amount of both legal and banned pesticides associated with them raises serious concerns about human safety, environmental damage, degradation of public lands, and poisoning of wildlife.

Our grow-site distribution model can help quantify the population-level impacts of these activities on natural resources, including to three forest species of high conservation concern.

The results can also be used by law enforcement and resource managers to prioritize and make more efficient their efforts to discover and remediate trespass grow sites, although it should be updated to account for changing environmental conditions and grower selection criteria. The map could be made dynamic, and the results mapped in various ways to facilitate decision-making. For example, we illustrated how the model results could be summarized and mapped at the subwatershed scale (Fig 3), which seems a useful resolution for planning aerial reconnaissance efforts or conservation and management actions. The results could also be summarized at any other scale or analytical unit of interest, such as whole watersheds, counties, or national forests. Regardless, it seems clear that a significant investment is needed to address this extensive threat to sensitive species, water quality, and other resource values.

## Acknowledgments

The authors would like to thank the following individuals for tireless assistance with ground-truthing: IERC field crew including C. Kamoroff, N. Soderfelt, K. Burton, S. Von Arb; California Department of Fish and Wildlife Game Wardens, B. Lynch, A. Galwey, and S. Crowl; US Forest Service Law Enforcement Officers, J. Mack and C. Magallon; Hoopa Tribal Police Officer, R. Buckman; Hoopa Tribal Wildlife Crew, A. Pole, C. Goddard, and D. Blake; CA Waterboards staff member, C. McIntee; Thanks to the following who assisted with considerations, design, analysis, and interpretation of the model: R. Davis, J. Dunk, W. Zielinski, S. Frick, D. Van Dyke, and E. Hinkley. Thank you to United States Drug Enforcement Agency's Domestic Cannabis Eradication and Suppression Program, California Department of Justice's Campaign Against Marijuana Planting and North State Marijuana Enforcement Team, and United States Forest Service Law Enforcement and Investigations for providing the cultivation site locality data and logistical support.

## Author Contributions

**Conceptualization:** Greta M. Wengert, J. Mark Higley, Mourad W. Gabriel.

**Data curation:** Mourad W. Gabriel, Craig Thompson.

**Formal analysis:** Greta M. Wengert, J. Mark Higley, Heather Rustigian-Romsos, Wayne D. Spencer.

**Funding acquisition:** Greta M. Wengert, Mourad W. Gabriel, Wayne D. Spencer, Deana L. Clifford.

**Investigation:** Greta M. Wengert, J. Mark Higley, Mourad W. Gabriel, Heather Rustigian-Romsos, Wayne D. Spencer, Deana L. Clifford, Craig Thompson.

**Methodology:** Greta M. Wengert, J. Mark Higley, Mourad W. Gabriel, Heather Rustigian-Romsos, Wayne D. Spencer, Deana L. Clifford, Craig Thompson.

**Project administration:** Greta M. Wengert, Deana L. Clifford.

**Resources:** Greta M. Wengert, Mourad W. Gabriel, Wayne D. Spencer, Deana L. Clifford.

**Software:** Greta M. Wengert, J. Mark Higley, Heather Rustigian-Romsos, Wayne D. Spencer.

**Supervision:** Greta M. Wengert, Heather Rustigian-Romsos, Wayne D. Spencer.

**Validation:** Greta M. Wengert, J. Mark Higley, Heather Rustigian-Romsos.

**Writing – original draft:** Greta M. Wengert.

**Writing – review & editing:** Greta M. Wengert, J. Mark Higley, Mourad W. Gabriel, Heather Rustigian-Romsos, Wayne D. Spencer, Deana L. Clifford, Craig Thompson.

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
