## [Decision Letter · Decision Letter 0]

7 Apr 2021

PONE-D-20-38479

Distribution of Trespass Cannabis Cultivation and Its Risk to Sensitive Forest Predators in California and Southern Oregon

PLOS ONE

Dear Dr. Wengert,

Thank you for submitting your manuscript to PLOS ONE. After careful consideration, we feel that it has merit but does not fully meet PLOS ONE’s publication criteria as it currently stands. Therefore, we invite you to submit a revised version of the manuscript that addresses the points raised during the review process.

We look forward to receiving your revised manuscript.

Kind regards,

Ji-Zhong Wan

Academic Editor

PLOS ONE

Journal Requirements:

'..We thank those contributing funding to this project: United States Fish and Wildlife Service, Yreka Field Office; California Department of Fish and Wildlife, US Fish and Wildlife Service Endangered Species Conservation and Recovery Grant; USDA Forest Service, Region 5; USDA Forest Service Law Enforcement and Investigations...'

'This study was funded by United States Fish and Wildlife Service, Yreka Field Office, through a grant administered by National Fish and Wildlife Foundation, grant #0206.15.050267, and funding from California Department of Fish and Wildlife, Endangered Species Act Traditional Section 6 grant #P1482006.The funders had no role in study design, data collection and analysis, decision to publish, or preparation of the manuscript.'

4. We note that Figures 1, 2, 3 and 5 in your submission contain map images which may be copyrighted.

a. You may seek permission from the original copyright holder of Figures 1, 2, 3 and 5 to publish the content specifically under the CC BY 4.0 license. 

5. Please upload a new copy of Figure 4 as the detail is not clear.

Please follow the link for more information: https://blogs.plos.org/plos/2019/06/looking-good-tips-for-creating-your-plos-figures-graphics/

Additional Editor Comments:

Pleases revised the mansucript fully according to the reviewers' comments. It is possible that the revised manuscript will be re-reviewed before publication.

Reviewers' comments:

Reviewer's Responses to Questions

**Comments to the Author**

1. Is the manuscript technically sound, and do the data support the conclusions?

Reviewer #1: Partly

Reviewer #2: Yes

2. Has the statistical analysis been performed appropriately and rigorously? 

Reviewer #1: I Don't Know

Reviewer #2: Yes

3. Have the authors made all data underlying the findings in their manuscript fully available?

Reviewer #1: No

Reviewer #2: Yes

4. Is the manuscript presented in an intelligible fashion and written in standard English?

Reviewer #1: Yes

Reviewer #2: Yes

5. Review Comments to the Author

Reviewer #1: The study is interesting and original, linking a problem of law enforcement with a problem of wildlife conservation. The map models seem well done with in particular a validation in the field which reinforces the results obtained for the possible presence of cannabis (although I am not a specialist in this question and therefore able to accurately criticize this aspect of the work).

I suggest checking and applying Plos One's criteria for writing, especially for citations.

The study draws a model-based parallel between the presence of a cannabis field and the presence of three protected predators. It begins with the fact that studies have shown that the use of rodenticides seriously affects those predators. However, the study does not provide direct proof of the presence of rodenticides and of their effect on the presence or absence of the predators potentially affected by cannabis plantation. In this way the conclusions offer a perspective about the possibility cannabis trespass cultivation and the possibility of predators' affectation. However, the study does offer the interest of asking, by obtaining modeled distributions, an interesting question: do illegal cannabis camps have an impact on wildlife, due to the cross-checking of the maps and the known effect of rodenticides on predators. This conclusion which is rather an open question offers interesting perspectives but requires more work and future research to ensure that there is really a link between these cannabis cultivation, the presence of rodenticides and the effects on predators. So, I think the work should be reformulated to be published which in my perspective warrants a major review of the work before it can be published.

The introduction could be improved, the authors begin by placing a lot of emphasis on the effect of rodenticides on predators, while this information, even if essential for giving perspectives to work, is not central. Also, this information could be better used in the discussion to justify the perspectives opened by the study. Also, I suggest instead formulating an introduction based on 1) the problem and necessity of researching possible cannabis field distribution, 2) the distribution of 3 predators which is likely to coincide with that of cannabis fields 3) the biology of these predators, in particular their diet and their habitats and to formulate hypotheses on 1) the need to model and know these distributions and 2)if those distributions coincide/overlaps or not.

In discussion it will then be possible to discuss these distributions and their overlaps, with possible effects of rodenticides (according to previous studies) and to formulate an opening on future studies as well as discuss the possible need to join efforts for law enforcement and wildlife protection.

Reviewer #2: In the proposed manuscript, the authors investigated the impact if the illegal cannabis cultivation performed on public lands on wildlife habitats in California and southern Oregon. The addressed problem is valid and the importance of the problem is significant. The depth and scale of this project is very impressive. The model, the methods, the data collection, data analysis and interpretation are appropriate. The results are properly introduced and discussed. The paper is well written.

6. PLOS authors have the option to publish the peer review history of their article (what does this mean?). If published, this will include your full peer review and any attached files.

Reviewer #1: No

Reviewer #2: No

---

## [Author Response · Author response to Decision Letter 0]

3 Jun 2021

Dear Editors,

I have made substantial changes to this manuscript (“Distribution of trespass cannabis cultivation and its risk to sensitive forest predators in California and Southern Oregon”) in direct response to the Academic Editor’s and Reviewers’ comments. A concern of one of the reviewers was the focus on anticoagulant rodenticides and their presence at grow sites and subsequent risk to predators interacting with these sites. I added several citations and reports that directly document the presence of anticoagulant rodenticides and other pesticides at these sites. The manuscript already had several citations directly showing the mortality and exposure risk to 2 of the 3 species covered in this study, so I reinforced the link by highlighting the actual findings of these papers. I also completely reworked the Introduction at the suggestion of Reviewer 1, to focus less on the effects and presence of rodenticides, and more on the need to model cannabis distribution in relation to the ecology of the three species we focused on.

The Academic Editor inquired about the ability to share the data we used to build the analysis and model. The entire dataset of locations we used is law enforcement sensitive. Every grow site location is still considered a crime scene still under investigation by either California Department of Justice or United States Forest Service Law Enforcement and Investigations, and thus, we are not allowed to share these locations outside of law enforcement agencies. This is for public safety reasons and also to protect the investigations. We are legally bound within the agreements we have with different law enforcement agencies to not disclose these data outside of our organization (Integral Ecology Research Center). 

I have also removed the reference to funding from the Acknowledgements section, and added a reference for logistical support on the part of the Forest Service. I have reworked the References section and the headings to match the formatting guidelines provided in the review.

I have not changed Figures 1, 2, 3, and 5 because they are not copyrighted. One of the authors (Rustigian-Romsos) created all those figures from publicly available spatial layers which are included at the end of this letter.

Finally, I have created a higher quality version of Figure 4 to meet quality requirements.

I believe we have adequately addressed all the concerns raised by Reviewer 1 and the Academic Editor, and that it is an improved manuscript.

Thank you for the opportunity to revise this work and prepare it for publication.

Sincerely,

Greta Wengert

Figure 1:

>Cities: from Populated Places, US Dept of transportation Bureau of Transportation Statistics,

https://data-usdot.opendata.arcgis.com/datasets/populated-places?geometry=-125.980%2C28.631%2C128.200%2C67.075

There are no access and use constraints

>States: from TIGER https://www.census.gov/geographies/mapping-files/time-series/geo/carto-boundary-file.html

Copyright protection is not available for any work of the United States Government (Title 17 U.S.C., Section 105).

https://www2.census.gov/geo/pdfs/maps-data/data/tiger/tgrshp2019/TGRSHP2019_TechDoc.pdf

>Hillshade: 3DEP Elevation Hillshade from USGS National Map https://www.usgs.gov/news/new-elevation-map-service-available-usgs-3d-elevation-program

>Public lands: PAD US - no use constraints, in the public domain:

https://databasin.org/datasets/df257819684049f9ad5f3b21a2a05e03/

https://databasin.org/datasets/99374897882a47499a0bc306fd4386b9/

https://databasin.org/datasets/fddafbb85f68418f8d9ff472d6ff25a8/

>Our modeling extent

Figure 2:

>Cities: from Populated Places, US Dept of transportation Bureau of Transportation Statistics,

https://data-usdot.opendata.arcgis.com/datasets/populated-places?geometry=-125.980%2C28.631%2C128.200%2C67.075

There are no access and use constraints

>States: from TIGER https://www.census.gov/geographies/mapping-files/time-series/geo/carto-boundary-file.html

Copyright protection is not available for any work of the United States Government (Title 17 U.S.C., Section 105).

https://www2.census.gov/geo/pdfs/maps-data/data/tiger/tgrshp2019/TGRSHP2019_TechDoc.pdf

>Hillshade: 3DEP Elevation Hillshade from USGS National Map https://www.usgs.gov/news/new-elevation-map-service-available-usgs-3d-elevation-program

>Our modeling extent 

>Our model output

Figure 3:

>Cities: from Populated Places, US Dept of transportation Bureau of Transportation Statistics,

https://data-usdot.opendata.arcgis.com/datasets/populated-places?geometry=-125.980%2C28.631%2C128.200%2C67.075

There are no access and use constraints

>States: from TIGER https://www.census.gov/geographies/mapping-files/time-series/geo/carto-boundary-file.html

Copyright protection is not available for any work of the United States Government (Title 17 U.S.C., Section 105).

https://www2.census.gov/geo/pdfs/maps-data/data/tiger/tgrshp2019/TGRSHP2019_TechDoc.pdf

>Hillshade: 3DEP Elevation Hillshade from USGS National Map https://www.usgs.gov/news/new-elevation-map-service-available-usgs-3d-elevation-program

>Our modeling extent 

>Our model output summarized by Hydrologic Units: Watershed Boundary Dataset (WBD) https://apps.nationalmap.gov/datasets/

Figure 5:

>States: from TIGER https://www.census.gov/geographies/mapping-files/time-series/geo/carto-boundary-file.html

Copyright protection is not available for any work of the United States Government (Title 17 U.S.C., Section 105).

https://www2.census.gov/geo/pdfs/maps-data/data/tiger/tgrshp2019/TGRSHP2019_TechDoc.pdf

>Hillshade: 3DEP Elevation Hillshade from USGS National Map https://www.usgs.gov/news/new-elevation-map-service-available-usgs-3d-elevation-program

>Our modeling extent

>Our model output

>Fisher habitat: US Fish and Wildlife Service 2015. We have this up on Data Basin under Creative Commons Attribution 3.0 License. 

https://databasin.org/datasets/38c1bc11ec4f4f8a935e7163ddfae58d/

>NSO habitat: Davis et al. 2016

>Humboldt marten habitat: Schrott and Shinn 2020

---

## [Decision Letter · Decision Letter 1]

4 Aug 2021

Distribution of trespass cannabis cultivation and its risk to sensitive forest predators in California and Southern Oregon

PONE-D-20-38479R1

Dear Dr. Wengert,

We’re pleased to inform you that your manuscript has been judged scientifically suitable for publication and will be formally accepted for publication once it meets all outstanding technical requirements.

Kind regards,

Ji-Zhong Wan

Academic Editor

PLOS ONE

Additional Editor Comments (optional):

Reviewers' comments:

Reviewer's Responses to Questions

**Comments to the Author**

1. If the authors have adequately addressed your comments raised in a previous round of review and you feel that this manuscript is now acceptable for publication, you may indicate that here to bypass the “Comments to the Author” section, enter your conflict of interest statement in the “Confidential to Editor” section, and submit your "Accept" recommendation.

Reviewer #1: All comments have been addressed

2. Is the manuscript technically sound, and do the data support the conclusions?

Reviewer #1: Yes

3. Has the statistical analysis been performed appropriately and rigorously? 

Reviewer #1: Yes

4. Have the authors made all data underlying the findings in their manuscript fully available?

Reviewer #1: No

5. Is the manuscript presented in an intelligible fashion and written in standard English?

Reviewer #1: Yes

6. Review Comments to the Author

Reviewer #1: The authors responded my concerns 1) the introduction is rewritten completely and 2) the author added information about mortality and exposure risk. I consider the article is ready for publication.

7. PLOS authors have the option to publish the peer review history of their article (what does this mean?). If published, this will include your full peer review and any attached files.

Reviewer #1: No

---

## [Editor Report · Acceptance letter]

9 Aug 2021

PONE-D-20-38479R1 

Distribution of trespass cannabis cultivation and its risk to sensitive forest predators in California and Southern Oregon 

Dear Dr. Wengert:

I'm pleased to inform you that your manuscript has been deemed suitable for publication in PLOS ONE. Congratulations! Your manuscript is now with our production department. 

Kind regards, 

on behalf of

Dr. Ji-Zhong Wan 

Academic Editor

PLOS ONE